# Comparison between SARS-CoV-2-Associated Acute Disseminated Encephalomyelitis and Acute Stroke: A Case Report

**DOI:** 10.3390/reports7010018

**Published:** 2024-03-01

**Authors:** Yu-Xuan Jiang, Ming-Hua Chen, Yen-Yue Lin, Yung-Hsi Kao, Ting-Wei Liao, Chih-Chien Chiu, Po-Jen Hsiao

**Affiliations:** 1Department of Emergency Medicine, Taoyuan Armed Forces General Hospital, Taoyuan 325, Taiwan; a0952591518@gmail.com (Y.-X.J.); yyline.tw@yahoo.com.tw (Y.-Y.L.); 2Division of Neurology, Department of Internal Medicine, Taoyuan Armed Forces General Hospital, Taoyuan 325, Taiwan; son0715@aftygh.gov.tw; 3Department of Emergency Medicine, Tri-Service General Hospital, National Defense Medical Center, Taipei 114, Taiwan; 4Department of Life Sciences, National Central University, Taoyuan 320, Taiwan; ykao@cc.ncu.edu.tw; 5Department of Internal Medicine, Taoyuan Armed Forces General Hospital, Taoyuan 325, Taiwan; doc62388@aftygh.gov.tw; 6Division of Infectious Disease, Department of Internal Medicine, Taoyuan Armed Forces General Hospital, Taoyuan 325, Taiwan; calebchiu.tw@gmail.com; 7Division of Infectious Disease, Department of Internal Medicine, Tri-Service General Hospital, National Defense Medical Center, Taipei 114, Taiwan; 8Division of Nephrology, Department of Internal Medicine, Taoyuan Armed Forces General Hospital, Taoyuan 325, Taiwan; 9Division of Nephrology, Department of Internal Medicine, Tri-Service General Hospital, National Defense Medical Center, Taipei 114, Taiwan

**Keywords:** acute disseminated encephalomyelitis, demyelinating, stroke, SARS-CoV-2, COVID-19

## Abstract

The neurological manifestations of severe acute respiratory syndrome coronavirus 2 (SARS-CoV-2) infection are underrecognized. Ischemic stroke and thrombotic complications have been documented in patients with SARS-CoV-2 infection. Acute disseminated encephalomyelitis (ADEM) associated with coronavirus disease 2019 (COVID-19) is rare but can occur; the incidence of COVID-19-associated ADEM is still not clear due to the lack of reporting of cases. ADEM may have atypical stroke-like manifestations, such as hemiparesis, hemiparesthesia and dysarthria. The treatment strategies for ADEM and acute stroke are different. Early identification and prompt management may prevent further potentially life-threatening complications. We report a patient with SARS-CoV-2 infection presenting with stroke-like manifestations. We also make a comparison between demyelinating diseases, COVID-19-associated ADEM and acute stroke. This case can prompt physicians to learn about the clinical manifestations of SARS-CoV-2-associated ADEM.

## 1. Introduction

Respiratory illness with severe acute respiratory syndrome coronavirus 2 (SARS-CoV-2) infection is most commonly described, and some neurological manifestations may occur. A systematic review and meta-analysis indicated that more than one-third of COVID-19 patients exhibited at least one neurological manifestation, including fatigue, headache, dizziness, disturbance of consciousness and so on [1].

Neurological complications may be caused by direct viral effects on neurons and glial cells, the immune-mediated response to virus and a hypercoagulable state, which may lead to systemic disease with potential life-threatening complications, such as acute stroke and demyelination [2,3,4].

Demyelinating diseases caused by SARS-CoV-2 are rare but include acute disseminated encephalomyelitis (ADEM), multiple sclerosis, and neuromyelitis optica [3,5]. ADEM is defined as acute and fulminant encephalopathy with multifocal neurologic findings, typically following a prodromal viral illness. ADEM is a diagnosis of exclusion and should be differentiated from other demyelinating diseases through cerebrospinal fluid (CSF) oligoclonal band analysis, brain magnetic resonance imaging (MRI) and some biomarkers [4]. Several reports have described a possible association between ADEM and SARS-CoV-2 infection [4,6,7,8]. Except COVID-19 infection, COVID-19 vaccinations have been reported to induce severe neurological adverse effects, such as ADEM, cerebrovascular stroke, cerebral venous sinus thrombosis and encephalitis [9,10,11,12]. However, the neuropathological mechanism of COVID-19-associated ADEM or ADEM after COVID-19 vaccinations is still not clear [4,13]. Similarly, the case presentation is a patient with SARS-CoV-2 infection presenting stroke-like manifestations, such as right hemiparesis, hemiparesthesia and dysarthria. In addition, we make a comparison between COVID-19-associated ADEM and acute stroke. This case may provide an understanding of the different manifestations between SARS-CoV-2-associated ADEM and acute stroke to physicians.

## 2. Detailed Case Description

In January 2023, a healthy 19-year-old man (height 167 cm, weight 47 cm, BMI 16.8 kg/m^2^) presented to our emergency department with right hemiparesis, hemiparesthesia and dysarthria 8 days after coronavirus disease 2019 (COVID-19) infection. Initially, he had only a slight respiratory illness, including symptoms such as cough and runny nose. Fever, headache and drowsiness developed gradually. He performed an at-home COVID-19 rapid antigen test on day 4 after the upper respiratory infection symptoms had begun, and the test confirmed COVID-19 infection. However, the neurological symptoms of slurred speech, weakness and numbness of the right upper and lower limbs came on suddenly after he took a nap in the afternoon 4 days later. There was no known neurological history and no family history of neurological disorders such as cerebrovascular accident, amyotrophic lateral sclerosis and multiple sclerosis. He received two doses of BNT-162b2 vaccination without obvious side effects 1 year prior.

On arrival, his initial vital signs were as follows: temperature, 37.4 °C; heart rate, 85 beats per minute; respiratory rate, 18 breaths per minute; blood pressure, 142/92 mmHg; and pulse oximetry, 97% on room air. He was lethargic, and the neurological examination (Figure 1) showed slurred speech and right upper and lower limb weakness in the absence of other neurologic signs (altered mental status, facial droop, aphasia, disorientation, gaze palsy, visual deficit, ataxia and inattention) and decreased muscle power of approximately three to four points. Weakness of the left upper and lower limbs was also reported, but the muscle power was normal. Although the patient had right upper and lower limb numbness, the pinprick test revealed no sensory loss. The score of the National Institutes of Health Stroke Scale was 3. The results of the complete blood count and blood biochemistry tests showed no significant abnormalities. A chest X-ray and electrocardiography also showed no obvious abnormalities, such as widened mediastinum or atrial fibrillation. A brain computed tomography (CT) revealed no focal hypodense lesions or intracranial hemorrhage.

Dual antiplatelet therapy (DAPT) with 300 mg aspirin and 300 mg clopidogrel was administered because this case was highly suspicious for presumed stroke in a patient with no known risk factors. Then, the patient was admitted for a workup.

After admission, the blood test results (Table 1), including autoimmune panels (ANA, anti-ds DNA, C3, C4, cANCA, pANCA and rheumatoid factor) and hypercoagulable states tests (protein C, protein S, antithrombin III, antiphospholipid antibody), were all negative. A lumbar puncture was performed, and the CSF analysis (Table 2) presented no pleocytosis, a normal total protein level of 30.5 (normal range: 15–45) mg/dL and a normal IgG index of 0.62 (normal range: 0–0.7). The CSF results showed negative findings for viral infections, bacteria, fungi and mycobacterium tuberculosis complex cultures. A repeated brain CT on admission day 2 still revealed no focal hypodense regions. A brain MRI demonstrated bilateral subcortical white matter and corpus callosum hyperintensity signal lesions on T2-weighted and fluid-attenuated inversion recovery (FLAIR). The demyelinating lesions also showed increased diffusivity on diffusion-weighted imaging (DWI) and decreased apparent diffusion coefficient (ADC) values (Figure 2). No evidence of large vessel occlusion or vascular territory involvement was noted.

The final diagnosis of COVID-19-associated ADEM was made according to the clinical features and MRI image findings. We then discontinued the DAPT prescription immediately. Medical treatment with a 3-day course of intravenous dexamethasone (10 mg/day) and a 3-day course of intravenous remdesivir (200 mg loading dose, then 100 mg/day) was administered. Although we tried relatively low-dose intravenous glucocorticoid therapy, the patient had a dramatically good response after 2 days. We did not titrate up to the high-dose glucocorticoid regimen because our treatment achieved a clinical response, and possible side effects were a concern. The patient had significant improvement after glucocorticoid and antiviral therapy during the hospital stay.

The patient was discharged and had a 7-day course of oral prednisolone 40 mg, a 5-day course of oral prednisolone 20 mg and then a 5-day course of oral prednisolone 10 mg. He fully recovered after a 3-month follow-up. Figure 3 shows the timeline of the patient from the day of COVID-19 infection and symptom onset to the 3-month follow-up.

## 3. Discussion

ADEM is an uncommon central nervous system (CNS) inflammatory demyelinating disease commonly associated with preceding viral infections. The duration from viral infection symptom onset to the development of ADEM varies from days to 6 weeks, with the majority occurring within 15–30 days in COVID-19 patients [7,8]. Our patient developed neurological deficits approximately one week after the onset of COVID-19 symptoms. The estimated annual incidence is approximately 1 to 3 per million in children [14]. ADEM is a very rare illness in adults, and the precise incidence is still unknown [4,14]. COVID-19-associated ADEM has been reported, but the incidence is higher in adults than in children [7,8]. ADEM is characterized by fulminant encephalopathy or neurological deficits. The typical presentations usually involve coma, paraparesis, quadriparesis, oculomotor deficits and dysarthria [7,8,14]. A brain MRI typically shows bilateral and asymmetric T2-weighted and FLAIR hyperintense lesions in the subcortical, periventricular white matter and deep gray matter, including the basal ganglia and thalamus [15]. In the acute stage, the MRI images usually demonstrate increased diffusivity on DWI and decreased ADC values [16]. The diagnostic guidelines for pediatric patients are based upon the clinical and radiologic features that were proposed by the International Pediatric Multiple Sclerosis Study Group [17]. The two major features are a first multifocal CNS demyelinating disease and encephalopathy, accompanied by an abnormal brain MRI during the acute phase. It is challenging to make a diagnosis of ADEM in adults due to the lack of established diagnostic criteria and a distinctive biological marker for adults. Encephalopathy is still an unclear diagnostic feature in adults. A retrospective study reported that only 56% of adult ADEM cases presented encephalopathy [18]. In our case, the typical features of encephalopathy, such as altered consciousness, confusion and irritability, were absent. Lethargy was considered to be a symptom of COVID-19 infection rather than encephalopathy. Acute onset neurological deficits, such as right hemiparesis, hemiparesthesia and dysarthria, were presented. It is intriguing that the brain MRI revealed bilateral demyelinating lesions that predominantly influenced the limbs on one side of the body. The treatment of ADEM includes supportive care and immune modulation therapy. Administration of high-dose glucocorticoids with a regimen of intravenous methylprednisolone at a dose of 1000 mg/day for 3 to 5 days is the first-line therapy. To reduce the risk of relapses, oral corticosteroid therapy should be continued and tapered gradually over 6 weeks [19]. Concurrent, empiric antibiotics and antiviral drugs can be considered. Intravenous immunoglobulin treatment (0.4 gm/kg/day for a 5-day course) and plasma exchange are options in patients who have a poor response to glucocorticoids or in some severe cases [4,6,12,19].

Ischemic stroke is also one of the most serious neurological complications in COVID-19 patients [3,20]. Coagulation dysfunctions may be caused by vascular endothelial damage that is secondary to direct viral damage, hypoxemia and cytokine storm [3,21,22]. It can lead to thrombosis and a hypercoagulable state. An estimated incidence of 1.4% was reported in the study by Nannoni S et al. [20]. They pooled 1106 patients with COVID-19 complicated by acute cerebrovascular disease and suggested that individuals with COVID-19 who developed stroke were more likely to be older and have preexisting cardiovascular comorbidities. However, the patients with COVID-19 infection and acute cerebrovascular disease were younger (approximately 6 years) than those presenting with stroke without COVID-19 [20]. Furthermore, Oxley TJ et al. reported several cases of large-vessel stroke in young COVID-19 patients and indicated that younger COVID-19 patients were also at risk for acute cerebrovascular disease [23]. The clinical characteristics between COVID-19-associated ADEM and acute stroke are shown in Table 3. Antiplatelet therapy, intravenous thrombolysis and endovascular thrombectomy are therapies for ischemic stroke [21]. In our case, the patient had no potential risk of stroke, such as hypercoagulable states, thrombosis or cardioembolism. The brain MRI did not demonstrate vascular territory infarction. However, based on the initial history and symptoms of our case, COVID-19-associated acute stroke may be easily misdiagnosed in the emergency department.

Except for COVID-19-associated acute stroke, it is challenging to differentiate ADEM from various inflammatory and demyelinating disorders, such as multiple sclerosis (MS), myelin oligodendrocyte glycoprotein (MOG) antibody-associated disease and neuromyelitis optica spectrum disorder (NMOSD) (Table 4) [5]. Our case study is limited by the lack of a CSF oligoclonal band analysis, serum or CSF anti-MOG-IgG and anti-aquaporin 4 (AQP4) antibody studies, repeated brain MRI and a long-term follow up. Although our patient did not present optic neuritis or transverse myelitis, it would be beneficial to consider running tests for anti-MOG-IgG and serum AQP4 antibodies, which are potentially compatible with MOG-IgG-mediated disease or NMOSD. Oligoclonal bands in CSF may point to a diagnosis of multiple sclerosis, although they may be present in some ADEM patients [8,17]. A repeated MRI image and a long-term follow-up are appropriate to differentiate ADEM from a first attack of MS, CNS inflammatory disorders or other demyelinating diseases. However, our patient denied any further tests, including blood studies (anti-MOG-IgG and anti-AQP4 antibodies), repeated CSF analysis and brain MRI because his symptoms resolved completely after the treatment. The limits of our case study result in diagnostic uncertainty.

## 4. Conclusions

In conclusion, the clinical presentations of COVID-19-associated ADEM may mimic acute stroke. In adult COVID-19 patients, classic symptoms of ADEM, such as encephalopathy, may not be present. This can cause misdiagnosis or delay treatment. However, the treatment strategy between these two diseases is different. The treatment of stroke mainly involves anticoagulants, IV thrombolysis and endovascular thrombectomy. Considering the potential coagulation dysfunctions in COVID-19 patients, a misdiagnosis and corresponding treatments may increase the risks of cerebral hemorrhage. Early detection of ADEM and timely administration of an immune modulation therapy, such as IV steroids, may reduce permanent disability in these patients. Early identification and well-timed therapy may help to achieve favorable clinical outcomes.

## Figures and Tables

**Figure 1 reports-07-00018-f001:**
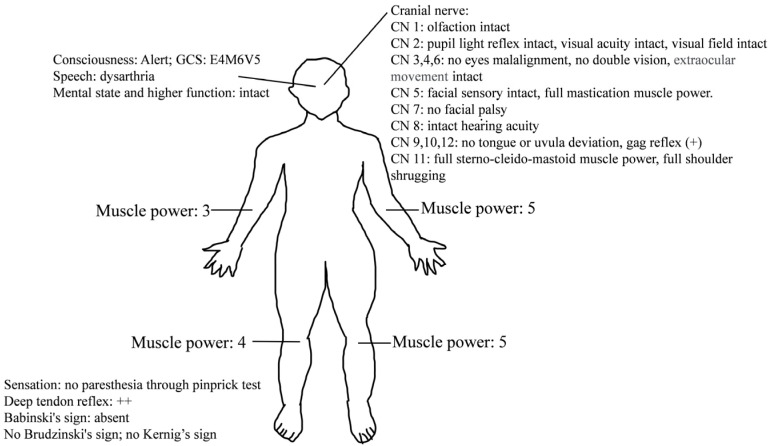
Neurological examinations of our patient.

**Figure 2 reports-07-00018-f002:**
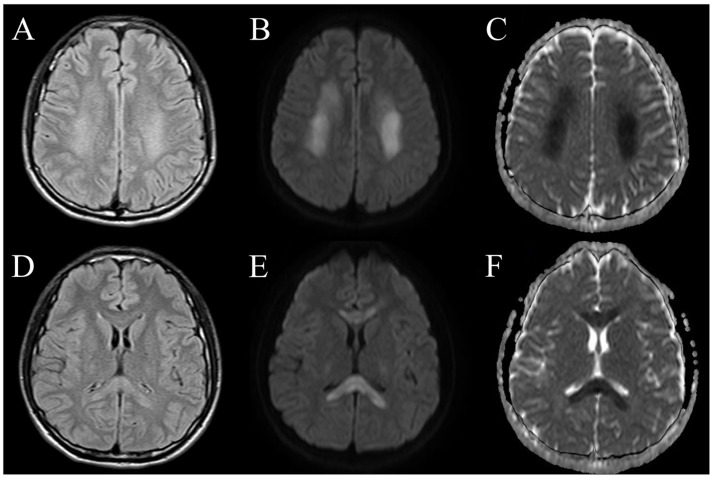
Axial brain MRI sequences of the patient with COVID-19-associated ADEM. Subcortical white matter lesions displayed bilateral diffuse, hyperintense signals on T2-weighted FLAIR images (**A**), increased diffusivity on DWI (**B**) and decreased ADC values (**C**). Corpus callosum lesions demonstrated bilateral T2-weighted FLAIR hyperintensities (**D**), increased diffusivity on DWI (**E**) and subtle restricted diffusion on ADC (**F**).

**Figure 3 reports-07-00018-f003:**
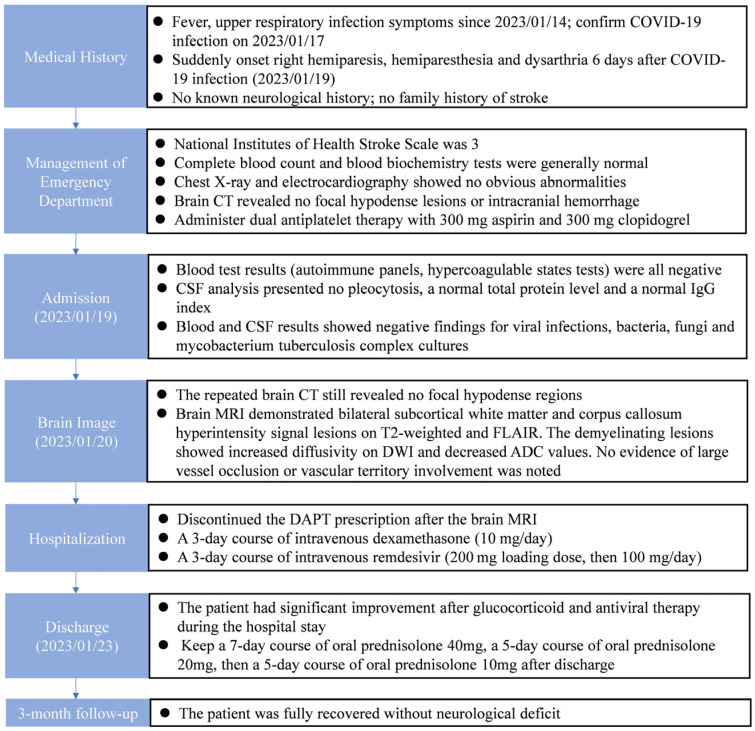
Timeline of the patient from the day of COVID-19 infection and symptom onset to the 3-month follow-up.

**Table 1 reports-07-00018-t001:** The results of blood examinations.

Item	Result	Unit	Normal Range
White blood cells	5280	/μL	4800–10,800
C-reactive protein	0.57	mg/dL	<0.5
Procalcitonin	<0.05	ng/mL	<0.07
Triglyceride	72.3	mg/dL	21–175
Total cholesterol	131.3	mg/dL	110–200
LDL cholesterol	84.9	mg/dL	<100
Prothrombin time	11.4	s	9.4–12
Activated partial thromboplastin time	31.2	s	25.3–32.3
ESR	21	mm/1 h	<29
ANA	Negative		
Anti-ds DNA	Negative		
Rheumatoid factor	<10	IU/mL	<14
cANCA	Negative		
pANCA	Negative		
C3	82.9	mg/dL	87–200
C4	36.5	mg/dL	19–52
Anti-cardiolipin IgM	Negative		
Anti-cardiolipin IgG	Negative		
Anti-phospholipid antibody IgG	Negative		
Anti-β2 glycoprotein IgG	Negative		
Protein C	75.5	%	70–140
Protein S	81.8	%	63.5–149
Antithrombin III	94.7	%	83–128
SARS-CoV-2 PCR	Positive, cycle threshold value 14		
Anti-HIV test	Nonreactive		
RPR	Nonreactive		
TPPA/TPHA test	Negative		
Anti-HCV	Nonreactive		
HBsAg	Nonreactive		
Herpes simplex virus 1 IgM	Negative		
Herpes simplex virus 2 IgM	Negative		
Cytomegalovirus IgM	Negative		
Epstein-Barr virus IgM	Negative		

Abbreviations: LDL, low-density lipoprotein; ESR, erythrocyte sedimentation rate; ANA, anti-nuclear antibody; anti-dsDNA, anti-double stranded deoxyribonucleic acid; ANCA, anti-neutrophil cytoplasmic antibody; Ig, immunoglobulin; SARS-CoV-2, severe acute respiratory syndrome coronavirus 2; HIV, human immunodeficiency virus; RPR, rapid plasma regain; TPPA, treponema pallidum particle agglutination; TPHA, treponema pallidum hemagglutination assay; HCV, hepatitis C virus; HBsAg, hepatitis B surface antigen.

**Table 2 reports-07-00018-t002:** The results of cerebrospinal fluid examinations.

Item	Result	Unit	Normal Range
pH	7.103		
White blood cells	<5	/μL	
Red blood cells	25	/μL	
Total protein	30.5	mg/dL	15–45
Glucose	57.39	mg/dL	40–70
LDH	25.5	U/L	
Chloride	127.9	mmol/L	
IgG index	0.62		0–0.7
Gram stain	No bacteria		
CSF culture	No bacteria		
Indian ink	Not found		
Acid-fast stain	Not found		
TB PCR DNA	Negative		
TB culture	Negative		
VDRL	Non-Reactive		
Cytomegalovirus PCR	Not detected		
Herpes simplex virus 1 PCR	Not detected		
Herpes simplex virus 2 PCR	Not detected		
Human herpesvirus 6 PCR	Not detected		
Human parechovirus PCR	Not detected		
Enterovirus PCR	Not detected		
Varicella zoster virus PCR	Not detected		
Cryptococcus neoformans/gattii PCR	Not detected		
Neisseria meningitidis	Not detected		
Listeria monocytogenes	Not detected		
Streptococcus agalactiae	Not detected		
Streptococcus pneumoniae	Not detected		
Escherichia coli K1	Not detected		
Haemophilus influenzae	Not detected		

Abbreviations: CSF, cerebrospinal fluid; LDH, lactate dehydrogenase; IgG, immunoglobulin G; TB, tuberculosis; PCR, polymerase chain reaction; DNA, deoxyribonucleic acid; VDRL, venereal disease research laboratory.

**Table 3 reports-07-00018-t003:** Comparison between COVID-19-associated ADEM and acute stroke.

Clinical Characteristics	COVID-19-Associated ADEM [6]	COVID-19-Associated Stroke [20]
Incidence	The incidence of classic ADEM is approximately 2–5 per million per year in children.	Pooled incidence of 1.4%
	However, the incidence of COVID-19-associated ADEM and ADEM in adults is not clear due to the lack of standardized reporting of cases.	
Age	Advanced age (nearly half are >50 years old)	Median 65.3 years
	In contrast to classic ADEM, COVID-19-associated ADEM occurs more in adults than children.	In comparison to stroke patients without COVID-19, people with COVID-19 and stroke were younger.
Duration since COVID-19 symptom onset	Usually occurring within 15–30 days	Median 8.8 days
Neurologic signs	Encephalopathy Focal motor deficits (paraparesis, quadriparesis) Cranial nerve deficits (oculomotor deficits, dysarthria) Focal sensory deficits Seizure Aphasia	Unilateral numbness or weakness of the face, arm or leg Aphasia Dysarthria Disorientation Ataxia Median NIHSS ^†^ 15
Radiological features	T2 FLAIR: diffuse, multifocal hyperintensities in the supratentorial and infratentorial white matter, but may also involve gray matter and/or the spinal cord. DWI: increased diffusivity ADC: decreased values in the acute stage; increased values in the subacute stage [16]. Some patients (42%) had evidence for hemorrhage on brain MRI, significantly higher than classic ADEM (2% in prior studies) [6].	Large vessel occlusion Multiple vascular territory infarction
Treatment	IV methylprednisolone IV immunoglobulin Plasmapheresis COVID-19-directed therapies	Antiplatelet therapy IV thrombolysis Endovascular thrombectomy
Prognosis	mRS ^‡^ score 6 (mortality): 20% mRS score 4–5 (severe disability): 20% mRS score 0–1 (no disability): 11%	In-hospital death: 31.5% Discharged to rehabilitation facilities: 25.7% Discharged home: 19.1%

Abbreviations: COVID-19, coronavirus disease 2019; ADEM, acute disseminated encephalomyelitis; T2 FLAIR, T2-weighted and fluid-attenuated inversion recovery; DWI, diffusion-weighted imaging; ADC, apparent diffusion coefficient; IV, intravenous. ^†^ The National Institutes of Health Stroke Scale (NIHSS) score ranges from 0 to 42, with higher numbers indicating more severe stroke. Stroke severity was higher in patients with COVID-19 (pooled median difference for NIHSS score 5). ^‡^ The range of modified Rankin Scale (mRS) scores is from 0 to 6, which measures the degree of disability or dependence in the daily activities of people who have suffered causes of neurological disability.

**Table 4 reports-07-00018-t004:** Differential diagnosis of acute disseminated encephalomyelitis.

	Acute Disseminated Encephalomyelitis	Multiple Sclerosis	Myelin Oligodendrocyte Glycoprotein Antibody-Associated Disease	Neuromyelitis Optica Spectrum Disorder
Clinical features	Acute and fulminant encephalopathy with multifocal neurologic findings; monophase; typically follows a prodromal viral illness	Chronic inflammation and demyelination; relapsing–remitting course; the multiphase; may not follow a prodromal viral illness	Central nervous system demyelination including ADEM, ON, TM; the most common is ON; monophasic or relapsing	ON, TM, area postrema syndrome; typically relapsing
Radiographic features	Poorly marginated lesions with larger bilateral but asymmetric white matter abnormalities in MRI	Ovoid plaques MRI lesions; hypointense T1-weighted lesions (black holes); Dawson fingers on sagittal views	ADEM-like MRI; enhancement of optic nerve MRI	Enhancement of optic nerve MRI
CSF analysis	Variable; nonspecific	Presence of oligoclonal bands; elevated proteins	Oligoclonal bands are typically absent; MOG-IgG autoantibody (+) in CSF	Variable; nonspecific
Serum autoantibodies	No specific biomarkers	No specific biomarkers	MOG-IgG autoantibody (+)	Anti-AQP4-IgG antibody (+)

Abbreviations: ADEM, acute disseminated encephalomyelitis; ON, optic neuritis; TM, transverse myelitis; MRI, magnetic resonance imaging; CSF, cerebrospinal fluid; MOG, myelin oligodendrocyte glycoprotein; IgG, immunoglobulin G; AQP4, aquaporin 4.

## Data Availability

The original contributions presented in this study are included in the article, and further inquiries can be directed to the corresponding author.

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
