# Peer review of "Comparison between SARS-CoV-2-Associated Acute Disseminated Encephalomyelitis and Acute Stroke: A Case Report"

_reports, 2024, doi:10.3390/reports7010018_

Round 1

Reviewer 1 Report

Comments and Suggestions for Authors

The case report is very interesting and brings a lot of insight for the clinicians. The abstract could be enhances with more valuable information. Also, I would like to see more informations about the distinct diseases that conjuncted in this case report, in the Introduction section. Pay attention when adding citations into the text. Also, add at least 2 figures and two tables. Add a chapter with the importance of fast diagnosis and add more information in the conclusion area. 

Author Response

Response to Reviewer 1

[General Comment]

The case report is very interesting and brings a lot of insight for the clinicians. The abstract could be enhances with more valuable information. Also, I would like to see more informations about the distinct diseases that conjuncted in this case report, in the Introduction section. Pay attention when adding citations into the text. Also, add at least 2 figures and two tables. Add a chapter with the importance of fast diagnosis and add more information in the conclusion area.

Author Reply: We sincerely appreciate your time and effort spent reviewing this manuscript. We have revised the manuscript thoroughly according to you and the reviewer’s suggestions.

Please see the section of abstract. (Page 1, line 26-37)

Abstract: The neurological manifestations of severe acute respiratory syndrome coronavirus 2 (SARS-CoV-2) infection are underrecognized. Ischemic stroke and thrombotic complications have been documented in patients with SARS-CoV-2 infection. Acute disseminated encephalomyelitis (ADEM) associated with coronavirus disease 2019 (COVID-19) is rare but can occur, the incidence of COVID-19-associated ADEM is still not clear due to the lack of reporting of cases. ADEM may have atypical stroke-like manifestations, such as hemiparesis, hemiparesthesia and dysarthria. The treatment strategies for ADEM and acute stroke are different. Early identification and prompt management may prevent further potentially life-threatening complications. We reported a patient with SARS-CoV-2 infection presenting with stroke-like manifestations. We also made a comparison between demyelinating diseases, COVID-19-associated ADEM and acute stroke. This case can prompt physicians to learn about the clinical manifestations of SARS-CoV-2-associated ADEM.

Please see the section of Introduction (Page 1, line 40- Page 2, line 66)

  1. Introduction

Respiratory illness with severe acute respiratory syndrome coronavirus 2 (SARS-CoV-2) infection is most commonly described, and some neurological manifestations may occur. A systematic review and meta-analysis indicated that more than one-third of COVID-19 patients exhibited at least one neurological manifestation including fatigue, headache, dizziness, disturbance of consciousness and son on.[1].

Neurological complications may be caused by direct viral effects on neurons and glial cells, the immune-mediated response to virus, and a hypercoagulable state, which may lead to systemic disease with potential life-threatening complications such as acute stroke and demyelination [2–4].

Demyelinating diseases caused by SARS-CoV-2 are rare but include acute disseminated encephalomyelitis (ADEM), multiple sclerosis, and neuromyelitis optica [3,5]. ADEM is defined as acute and fulminant encephalopathy with multifocal neurologic findings, typically follows a prodromal viral illness. ADEM is a diagnosis of exclusion and should be differentiated with other demyelinating diseases through cerebrospinal fluid (CSF) oligoclonal band analysis, brain magnetic resonance imaging (MRI) and some biomarkers [4]. Several reports have described a possible association between ADEM and SARS-CoV-2 infection [4,6-8]. Except COVID-19 infection, COVID-19 vaccinations induce severe neurological adverse effects such as ADEM [9], cerebrovascular stroke [10], cerebral venous sinus thrombosis [11] and encephalitis [12] have been reported. However, the neuropathological mechanism of COVID-19-associated ADEM or ADEM after COVID-19 vaccinations are still not clear [4,13]. Similarly, the case presentation is a patient with SARS-CoV-2 infection presenting stroke-like manifestations such as right hemiparesis, hemiparesthesia and dysarthria. In addition, we made a comparison between COVID-19-associated ADEM and acute stroke. This case may provide an understanding of the different manifestations between SARS-CoV-2-associated ADEM and acute stroke to the physicians.

We added more citations into the text and check that all references are relevant to the contents of the manuscript. (Page 10, line 263- Page 11, line 311)

References

  1. Misra S, Kolappa K, Prasad M, et al. Frequency of Neurologic Manifestations in COVID-19: A Systematic Review and Meta-analysis. Neurology. 2021, 97(23): e2269-e2281. doi: 10.1212/WNL.0000000000012930.
  2. Gavriilaki E, Anyfanti P, Gavriilaki M, et al. Endothelial Dysfunction in COVID-19: Lessons Learned from Coronaviruses. Curr Hypertens Rep. 2020, 22(9), 63. doi: 10.1007/s11906-020-01078-6.
  3. Dai X, Cao X, Jiang Q, et al. Neurological complications of COVID-19. QJM. 2023, 116(3): 161-180. doi: 10.1093/qjmed/hcac272.
  4. Stoian A, Bajko Z, Stoian M, et al. The Occurrence of Acute Disseminated Encephalomyelitis in SARS-CoV-2 Infection/Vaccination: Our Experience and a Systematic Review of the Literature. Vaccines (Basel). 2023, 11(7), 1225. doi: 10.3390/vaccines11071225.
  5. Feizi P, Sharma K, Pasham SR, et al. Central nervous system (CNS) inflammatory demyelinating diseases (IDDs) associated with COVID-19: A case series and review. J Neuroimmunol. 2022, 371, 577939. doi:10.1016/j.jneuroim.2022.577939.
  6. Zanin L, Saraceno G, Panciani PP, et al. SARS-CoV-2 can induce brain and spine demyelinating lesions. Acta Neurochir (Wien). 2020, 162(7), 1491-1494. doi: 10.1007/s00701-020-04374-x.
  7. Manzano GS, McEntire CRS, Martinez-Lage M, et al. Acute Disseminated Encephalomyelitis and Acute Hemorrhagic Leukoencephalitis Following COVID-19: Systematic Review and Meta-synthesis. Neurol Neuroimmunol Neuroinflamm. 2021, 8(6), 1080. doi: 10.1212/NXI.0000000000001080.
  8. Parsons T, Banks S, Bae C, et al. COVID-19-associated acute disseminated encephalomyelitis (ADEM). J Neurol. 2020, 267(10), 2799-2802. doi:10.1007/s00415-020-09951-9.
  9. Nabizadeh F, Noori M, Rahmani S, Hosseini H. Acute disseminated encephalomyelitis (ADEM) following COVID-19 vaccination: A systematic review. J Clin Neurosci. 2023, 111: 57-70. doi: 10.1016/j.jocn.2023.03.008.
  10. Kakovan M, Ghorbani Shirkouhi S, Zarei M, Andalib S. Stroke Associated with COVID-19 Vaccines. J Stroke Cerebrovasc Dis. 2022, 31(6): 106440. doi: 10.1016/j.jstrokecerebrovasdis.2022.106440.
  11. Wang RL, Chiang WF, Shyu HY, et al. COVID-19 vaccine-associated acute cerebral venous thrombosis and pulmonary artery embolism. QJM. 2021, 114(7): 506-507. doi: 10.1093/qjmed/hcab185.
  12. Fan HT, Lin YY, Chiang WF, et al. COVID-19 vaccine-induced encephalitis and status epilepticus. QJM. 2022, 115(2): 91-93. doi: 10.1093/qjmed/hcab335.
  13. Chen WP, Chen MH, Shang ST, et al. Investigation of Neurological Complications after COVID-19 Vaccination: Report of the Clinical Scenarios and Review of the Literature. Vaccines (Basel). 2023, 11(2): 425. doi: 10.3390/vaccines11020425. 
  14. Pohl D, Alper G, Van Haren K, et al. Acute disseminated encephalomyelitis: Updates on an inflammatory CNS syndrome. Neurology. 2016, 87(9 Suppl 2), S38-45. doi: 10.1212/WNL.0000000000002825.
  15. Bhawna S, Rahul H, Kadam N, et al. Transient splenial diffusion-weighted image restriction mimicking stroke. Am J Emerg Med. 2014, 32(9), 1156.e1-2. doi: 10.1016/j.ajem.2014.02.044.
  16. Balasubramanya KS, Kovoor JM, Jayakumar PN, et al. Diffusion-weighted imaging and proton MR spectroscopy in the characterization of acute disseminated encephalomyelitis. Neuroradiology. 2007, 49(2), 177-83. doi: 10.1007/s00234-006-0164-2.
  17. Krupp LB, Tardieu M, Amato MP, et al. International Pediatric Multiple Sclerosis Study Group criteria for pediatric multiple sclerosis and immune-mediated central nervous system demyelinating disorders: revisions to the 2007 definitions. Mult Scler. 2013, 19(10), 1261-1267. doi: 10.1177/1352458513484547.
  18. Koelman DL, Chahin S, Mar SS, et al. Acute disseminated encephalomyelitis in 228 patients: A retrospective, multicenter US study. Neurology. 2016, 86(22), 2085-2093. doi: 10.1212/WNL.0000000000002723.
  19. Alexander M, Murthy JM. Acute disseminated encephalomyelitis: Treatment guidelines. Ann Indian Acad Neurol. 2011, 14(Suppl 1), S60-43. doi: 10.4103/0972-2327.83095.
  20. Nannoni S, de Groot R, Bell S,et al. Stroke in COVID-19: A systematic review and meta-analysis. Int J Stroke. 2021, 16(2), 137-149. doi: 10.1177/1747493020972922.
  21. Kits A, Pantalone MR, Illies C, et al. Fatal Acute Hemorrhagic Encephalomyelitis and Antiphospholipid Antibodies following SARS-CoV-2 Vaccination: A Case Report. Vaccines (Basel). 2022, 10(12), 2046. doi: 10.3390/vaccines10122046.
  22. Asakura H, Ogawa H. COVID-19-associated coagulopathy and disseminated intravascular coagulation. Int J Hematol. 2021, 113(1): 45-57. doi: 10.1007/s12185-020-03029-y.
  23. Oxley TJ, Mocco J, Majidi S, et al. Large-Vessel Stroke as a Presenting Feature of Covid-19 in the Young. N Engl J Med. 2020, 382(20), e60. doi:10.1056/NEJMc2009787.

We add 2 figures and 3 tables.

(Page 2, line 82; Page 3, line 94)

He was lethargic, and the neurological examination (Figure 1) showed slurred speech and right upper and lower limb weakness in the absence of other neurologic signs (altered mental status, facial droop, aphasia, disorientation, gaze palsy, visual deficit, ataxia and inattention) and decreased muscle power of approximately 3 to 4 points.

(Page 5, line 143; Page 6, line 146)

Figure 3 shows the timeline of the patient from the day of COVID-19 infection and symptom onset to the 3-month follow-up.

(Page 3, line 100; Page 3, line 114)

After admission, the blood test results (Table 1), including autoimmune panels (ANA, anti-ds DNA, C3, C4, cANCA, pANCA and rheumatoid factor) and hypercoagulable states tests (protein C, protein S, antithrombin III, antiphospholipid antibody), were all negative.

(Page 3, line 103; Page 3, line 114)

Lumbar puncture was performed, and CSF analysis (Table 2) presented no pleocytosis, a normal total protein level of 30.5 (normal range: 15-45) mg/dL, and a normal IgG index of 0.62 (normal range: 0-0.7).

(Page 9, line 217; Page 9, line 231)

Except for COVID-19-associated acute stroke, it is challenging to differentiate ADEM from various inflammatory and demyelinating disorders, such as multiple sclerosis (MS), myelin oligodendrocyte glycoprotein (MOG) antibody-associated disease and neuromyelitis optica spectrum disorder (NMOSD) (Table 4) [5].

Please see the section of conclusions. (Page 10, line 235-245)

In conclusion, the clinical presentations of COVID-19-associated ADEM may mimic acute stroke. In adult COVID-19 patients, classic symptom of ADEM as encephalopathy may not present. This can make misdiagnosis or delay treatment. However, the treatment strategy between these two diseases is different. Treatment of stroke mainly involves anticoagulants, IV thrombolysis, and endovascular thrombectomy. Considering the potential coagulation dysfunctions in COVID-19 patient, a misdiagnosis and corresponding treatments may increase the risks of cerebral hemorrhage. Early detection of ADEM and timely administration of immune modulation therapy such as IV steroid may reduce permanent disability in these patients. Early identification and well-timed therapy may help to achieve favourable clinical outcomes.

Please see the attachment (Figures and tables)

Last, we are deeply honored by the time and effort you spent reviewing this manuscript. In reviewing and revising our manuscript, we are motivated to read more and thus learn more from your criticisms.

Reviewer 2 Report

Comments and Suggestions for Authors

The manuscript entitled„ Comparison between SARS-CoV-2-associated Acute Disseminated Encephalomyelitis and Acute Stroke: A Case Report presents in detail the case of a patient with a COVID-19 infection and ADEM comparing it with the clinical picture of an ischemic stroke.

The most significant shortcoming of the study is the doubtfulness of the scientific originality and novelty.

Moreover, there are already several systematic reviews related to Covid 19 infection and vaccination and ADEM as a complication (for example https://doi.org/10.1016/j.jocn.2023.03.008https://doi.org/10.1016/j.jocn.2023.03.008

Thus, the presented case does not bring anything particularly original from what has not already been written in the previous numerous case reports.

The literature on cases of acute disseminated encephalomyelitis (ADEM) associated with SARS-CoV-2 infection has been rapidly increasing, and the presented case is not particularly original in any way, which would distinguish it from the previous ones.

Author Response

Response to Reviewer 2

[General Comment]

The manuscript entitled„ Comparison between SARS-CoV-2-associated Acute Disseminated Encephalomyelitis and Acute Stroke: A Case Report presents in detail the case of a patient with a COVID-19 infection and ADEM comparing it with the clinical picture of an ischemic stroke.

The most significant shortcoming of the study is the doubtfulness of the scientific originality and novelty.

Moreover, there are already several systematic reviews related to Covid 19 infection and vaccination and ADEM as a complication (for example https://doi.org/10.1016/j.jocn.2023.03.008; https://doi.org/10.1016/j.jocn.2023.03.008.

Thus, the presented case does not bring anything particularly original from what has not already been written in the previous numerous case reports.

The literature on cases of acute disseminated encephalomyelitis (ADEM) associated with SARS-CoV-2 infection has been rapidly increasing, and the presented case is not particularly original in any way, which would distinguish it from the previous ones.

Author Reply: We sincerely appreciate your time and effort spent reviewing this manuscript. Although the cases of SARS-CoV-2 infection associated (ADEM) has been rapidly increasing. However, the neuropathological mechanism and incidence of COVID-19-associated ADEM or ADEM after COVID-19 vaccination are still not clear due to the lack of reporting of cases.

We present a case of SARS-CoV-2 infection presenting with stroke-like manifestations. In adult COVID-19 patients, classic symptom of ADEM as encephalopathy may not present. This can make misdiagnosis or delay treatment.   

Distinguish our case from the previous cases, we made a discuss about comparison between demyelinating diseases, COVID-19-associated ADEM and acute stroke. The treatment strategy between these two diseases is different. Treatment of stroke mainly involves anticoagulants, IV thrombolysis, and endovascular thrombectomy. Considering the potential coagulation dysfunctions in COVID-19 patient, a misdiagnosis and corresponding treatments may increase the risks of cerebral hemorrhage. Early detection of ADEM and timely administration of immune modulation therapy such as IV steroid may reduce permanent disability in these patients. This case can prompt an important neurological manifestation of COVID-19 to physicians.

We have revised the manuscript thoroughly according to you and the reviewer’s suggestions.

(Page 2, line 56-60)

Except COVID-19 infection, COVID-19 vaccinations induce severe neurological adverse effects such as ADEM, cerebrovascular stroke, cerebral venous sinus thrombosis and encephalitis have been reported [9-12]. However, the neuropathological mechanism of COVID-19-associated ADEM or ADEM after COVID-19 vaccinations are still not clear [4,13].

References

  1. Nabizadeh F, Noori M, Rahmani S, Hosseini H. Acute disseminated encephalomyelitis (ADEM) following COVID-19 vaccination: A systematic review. J Clin Neurosci. 2023, 111: 57-70. doi: 10.1016/j.jocn.2023.03.008.
  2. Kakovan M, Ghorbani Shirkouhi S, Zarei M, Andalib S. Stroke Associated with COVID-19 Vaccines. J Stroke Cerebrovasc Dis. 2022, 31(6): 106440. doi: 10.1016/j.jstrokecerebrovasdis.2022.106440.
  3. Wang RL, Chiang WF, Shyu HY, et al. COVID-19 vaccine-associated acute cerebral venous thrombosis and pulmonary artery embolism. QJM. 2021, 114(7): 506-507. doi: 10.1093/qjmed/hcab185.
  4. Fan HT, Lin YY, Chiang WF, et al. COVID-19 vaccine-induced encephalitis and status epilepticus. QJM. 2022, 115(2): 91-93. doi: 10.1093/qjmed/hcab335.
  5. Chen WP, Chen MH, Shang ST, et al. Investigation of Neurological Complications after COVID-19 Vaccination: Report of the Clinical Scenarios and Review of the Literature. Vaccines (Basel). 2023, 11(2): 425. doi: 10.3390/vaccines11020425. 

We added more specifics on the patient's initial COVID-19 illness severity and exact symptom onset timelines to enhance the clinical picture. Discussing limitations of our case such as lack of long-term follow up and diagnostic uncertainty. And added the importance of fast diagnosis in the conclusion area. 

(Page 2, line 74-75)

However, the neurological symptoms as slurred speech, weakness, numbness of right upper and lower limbs came on suddenly after he took a nap in the afternoon 4 days later.

(Page 2, line 82-88; Page 3, line 94)

On arrival, his initial vital signs were temperature, 37.4 °C; heart rate, 85 beats per minute; respiratory rate, 18 breaths per minute; blood pressure, 142/92 mmHg; and pulse oximetry, 97% on room air. He was lethargic, and the neurological examination (Figure 1) showed slurred speech and right upper and lower limb weakness in the absence of other neurologic signs (altered mental status, facial droop, aphasia, disorientation, gaze palsy, visual deficit, ataxia and inattention) and decreased muscle power of approximately 3 to 4 points. Weakness of the left upper and lower limbs was also reported, but the muscle power was normal. Although the patient had right upper and lower limbs numbness, the pinprick test revealed no sensory loss.

(Page 3, line 100; Page 3, line 114)

After admission, the blood test results (Table 1), including autoimmune panels (ANA, anti-ds DNA, C3, C4, cANCA, pANCA and rheumatoid factor) and hypercoagulable states tests (protein C, protein S, antithrombin III, antiphospholipid antibody), were all negative.

(Page 3, line 103; Page 3, line 114)

Lumbar puncture was performed, and CSF analysis (Table 2) presented no pleocytosis, a normal total protein level of 30.5 (normal range: 15-45) mg/dL, and a normal IgG index of 0.62 (normal range: 0-0.7).

(Page 5, line 143; Page 6, line 146)

Figure 3 shows the timeline of the patient from the day of COVID-19 infection and symptom onset to the 3-month follow-up.

Please see the section of discussion.

(Page 9, line 217-229)

Except for COVID-19-associated acute stroke, it is challenging to differentiate ADEM from various inflammatory and demyelinating disorders, such as multiple sclerosis (MS), myelin oligodendrocyte glycoprotein (MOG) antibody-associated disease and neuromyelitis optica spectrum disorder (NMOSD) (Table 4) [5]. Our case study is limited by the lack of CSF oligoclonal band analysis, serum or CSF anti- MOG-IgG and anti-aquaporin 4 (AQP4) antibody studies, repeated brain MRI and a long-term follow up. Although our patient did not present optic neuritis or transverse myelitis, it would be beneficial to consider running tests for anti-MOG-IgG and serum AQP4 antibodies, which are potentially compatible with MOG-IgG–mediated disease or NMOSD. Oligoclonal bands in CSF may point to a diagnosis of multiple sclerosis, although they may be present in some ADEM patients [8,17]. A repeated MRI image and a long-term follow-up are appropriate to differentiate ADEM from a first attack of MS, CNS inflammatory disorders or other demyelinating diseases. However, our patient denied any further tests, including blood studies (anti-MOG-IgG and anti-AQP4 antibodies), repeated CSF analysis and brain MRI, because his symptoms resolved completely after the treatment. The limits of our case study make diagnostic uncertainty.

Please see the section of conclusions. (Page 10, line 235-245)

In conclusion, the clinical presentations of COVID-19-associated ADEM may mimic acute stroke. In adult COVID-19 patients, classic symptom of ADEM as encephalopathy may not present. This can make misdiagnosis or delay treatment. However, the treatment strategy between these two diseases is different. Treatment of stroke mainly involves anticoagulants, IV thrombolysis, and endovascular thrombectomy. Considering the potential coagulation dysfunctions in COVID-19 patient, a misdiagnosis and corresponding treatments may increase the risks of cerebral hemorrhage. Early detection of ADEM and timely administration of immune modulation therapy such as IV steroid may reduce permanent disability in these patients. Early identification and well-timed therapy may help to achieve favourable clinical outcomes.

Please see the attachment (Tables and Figures)

Last, we are deeply honored by the time and effort you spent reviewing this manuscript. In reviewing and revising our manuscript, we are motivated to read more and thus learn more from your criticisms.

Reviewer 3 Report

Comments and Suggestions for Authors

Thank you for the opportunity to review this interesting case report manuscript. I read the report thoroughly and with great interest. The case highlights an important neurological manifestation of COVID-19.

The case presentation provides good clinical details about the patient's demographics, COVID-19 timeline, presenting symptoms, diagnostic workup, imaging findings, and treatment. The discussion makes insightful comparisons between ADEM and stroke. 

A few areas could potentially be expanded to add further details and depth. Including more specifics on the patient's initial COVID-19 illness severity and exact symptom onset timelines may enhance the clinical picture. Discussing limitations such as lack of long-term follow up and diagnostic uncertainty is advised.

In summary, this is an interesting case report describing COVID-19-associated ADEM with stroke-like features. 

Below are some suggestions for consideration:

Please consider deleting the first sentence Lines 26-27, or rephrase it as under-recorgnized or underreported.

Please consider providing more information such as incidence rates or percentage of occurrence in prior literature (Lines 28-29)

Please consider adding a numerical value to Lines 38-40. This would be more informative for the readers. 

Lines 40 to 43 consider a new paragraph for readers flow and to maintain clarity and avoiding losing readers interest 

Line 46, consider starting the sentence as follow: Similarly, the case presented is......

Consider breaking the paragraph into two under Section 2 detailed case description. perphaps second paraphgraph from Lines 63-73.

It would be essential to include a full neurological examination. An illustration would be helpful showing motor grades and sensory grades. Additionally, how the weakness progressed is important...since it may have been something gradual in onset compared to acute stroke.

Figure 1 footnote and images do not correlates. Please correct it.

Author Response

Response to Reviewer 3

[General Comment]

Thank you for the opportunity to review this interesting case report manuscript. I read the report thoroughly and with great interest. The case highlights an important neurological manifestation of COVID-19.

The case presentation provides good clinical details about the patient's demographics, COVID-19 timeline, presenting symptoms, diagnostic workup, imaging findings, and treatment. The discussion makes insightful comparisons between ADEM and stroke. 

A few areas could potentially be expanded to add further details and depth. Including more specifics on the patient's initial COVID-19 illness severity and exact symptom onset timelines may enhance the clinical picture. Discussing limitations such as lack of long-term follow up and diagnostic uncertainty is advised.

Author Reply: We sincerely appreciate your time and effort spent reviewing this manuscript. We have revised the manuscript thoroughly according to you and the reviewer’s suggestions.

Please see the section of Detailed Case Description, Table 1 and 2 and Figure 3. The responses to your comments are found below.

(Page 2, line 74-75)

However, the neurological symptoms as slurred speech, weakness, numbness of right upper and lower limbs came on suddenly after he took a nap in the afternoon 4 days later.

(Page 2, line 82-88)

On arrival, his initial vital signs were temperature, 37.4 °C; heart rate, 85 beats per minute; respiratory rate, 18 breaths per minute; blood pressure, 142/92 mmHg; and pulse oximetry, 97% on room air. He was lethargic, and the neurological examination (Figure 1) showed slurred speech and right upper and lower limb weakness in the absence of other neurologic signs (altered mental status, facial droop, aphasia, disorientation, gaze palsy, visual deficit, ataxia and inattention) and decreased muscle power of approximately 3 to 4 points. Weakness of the left upper and lower limbs was also reported, but the muscle power was normal. Although the patient had right upper and lower limbs numbness, the pinprick test revealed no sensory loss.

(Page 3, line 100-105)

After admission, the blood test results (Table 1), including autoimmune panels (ANA, anti-ds DNA, C3, C4, cANCA, pANCA and rheumatoid factor) and hypercoagulable states tests (protein C, protein S, antithrombin III, antiphospholipid antibody), were all negative. Lumbar puncture was performed, and CSF analysis (Table 2) presented no pleocytosis, a normal total protein level of 30.5 (normal range: 15-45) mg/dL, and a normal IgG index of 0.62 (normal range: 0-0.7).

(Page 5, line 143-144)

Figure 3 shows the timeline of the patient from the day of COVID-19 infection and symptom onset to the 3-month follow-up.

(Page 9, line 217-229)

Except for COVID-19-associated acute stroke, it is challenging to differentiate ADEM from various inflammatory and demyelinating disorders, such as multiple sclerosis (MS), myelin oligodendrocyte glycoprotein (MOG) antibody-associated disease and neuromyelitis optica spectrum disorder (NMOSD) (Table 4) [5]. Our case study is limited by the lack of CSF oligoclonal band analysis, serum or CSF anti- MOG-IgG and anti-aquaporin 4 (AQP4) antibody studies, repeated brain MRI and a long-term follow up. Although our patient did not present optic neuritis or transverse myelitis, it would be beneficial to consider running tests for anti-MOG-IgG and serum AQP4 antibodies, which are potentially compatible with MOG-IgG–mediated disease or NMOSD. Oligoclonal bands in CSF may point to a diagnosis of multiple sclerosis, although they may be present in some ADEM patients [8,17]. A repeated MRI image and a long-term follow-up are appropriate to differentiate ADEM from a first attack of MS, CNS inflammatory disorders or other demyelinating diseases. However, our patient denied any further tests, including blood studies (anti-MOG-IgG and anti-AQP4 antibodies), repeated CSF analysis and brain MRI, because his symptoms resolved completely after the treatment. The limits of our case study make diagnostic uncertainty.

In summary, this is an interesting case report describing COVID-19-associated ADEM with stroke-like features. 

Below are some suggestions for consideration:

Please consider deleting the first sentence Lines 26-27, or rephrase it as under-recorgnized or underreported.

Author Reply: Thank you for your valuable comments. We have made this correction. (Page 1, line 27)

The neurological manifestations of severe acute respiratory syndrome coronavirus 2 (SARS-CoV-2) infection are underrecognized.

Please consider providing more information such as incidence rates or percentage of occurrence in prior literature (Lines 28-29)

Author Reply: Thank you for your valuable comments. We have made this correction. (Page 1, line 29-30)

Acute disseminated encephalomyelitis (ADEM) associated with coronavirus disease 2019 (COVID-19) is rare but can occur, the incidence of COVID-19-associated ADEM is still not clear due to the lack of reporting of cases.

Please consider adding a numerical value to Lines 38-40. This would be more informative for the readers. 

Author Reply: Thank you for your valuable comments. We have made this correction. (Page 1, line 43-45)

A systematic review and meta-analysis indicated that more than one-third of COVID-19 patients exhibited at least one neurological manifestation including fatigue, headache, dizziness, disturbance of consciousness and son on.[1].

References

  1. Misra S, Kolappa K, Prasad M, et al. Frequency of Neurologic Manifestations in COVID-19: A Systematic Review and Meta-analysis. Neurology. 2021, 97(23): e2269-e2281. doi: 10.1212/WNL.0000000000012930.

Lines 40 to 43 consider a new paragraph for readers flow and to maintain clarity and avoiding losing readers interest 

Author Reply: Thank you for your valuable comments. We have made this correction. (Page 1, line 46-49)

Line 46, consider starting the sentence as follow: Similarly, the case presented is......

Author Reply: Thank you for your valuable comments. We have made this correction. (Page 2, line 61)

Similarly, the case presentation is a patient with SARS-CoV-2 infection presenting stroke-like manifestations such as right hemiparesis, hemiparesthesia and dysarthria.

Consider breaking the paragraph into two under Section 2 detailed case description. perphaps second paraphgraph from Lines 63-73.

Author Reply: Thank you for your valuable comments. We have made this correction. (Page 2, line 80-92)

It would be essential to include a full neurological examination. An illustration would be helpful showing motor grades and sensory grades. Additionally, how the weakness progressed is important...since it may have been something gradual in onset compared to acute stroke.

Author Reply: Thank you for your valuable comments. We have made this correction.

(Page 2, line 82-88)

On arrival, his initial vital signs were temperature, 37.4 °C; heart rate, 85 beats per minute; respiratory rate, 18 breaths per minute; blood pressure, 142/92 mmHg; and pulse oximetry, 97% on room air. He was lethargic, and the neurological examination (Figure 1) showed slurred speech and right upper and lower limb weakness in the absence of other neurologic signs (altered mental status, facial droop, aphasia, disorientation, gaze palsy, visual deficit, ataxia and inattention) and decreased muscle power of approximately 3 to 4 points. Weakness of the left upper and lower limbs was also reported, but the muscle power was normal. Although the patient had right upper and lower limbs numbness, the pinprick test revealed no sensory loss.

(Page 3, line 94)

Figure 1. Neurological examinations of our patient

Figure 1 footnote and images do not correlates. Please correct it.

Author Reply: Thank you for your valuable comments. We have made this correction. (Figure, Page 5, Line 126)

Please see the attachment. (Tables and Figures)

Last, we are deeply honored by the time and effort you spent reviewing this manuscript. In reviewing and revising our manuscript, we are motivated to read more and thus learn more from your criticisms.

Round 2

Reviewer 2 Report

Comments and Suggestions for Authors

The manuscript was thoroughly revised by the authors based on the feedback provided by the reviewers. As a result, the overall scientific quality of the manuscript has been considerably improved. The authors also added several tables and figures, which have significantly enhanced the quality of the manuscript. They have included relevant details in the text to clarify and emphasize the importance of early and accurate diagnosis of ADEM. Due to the improved writing, the manuscript is now suitable for publication and should be considered for acceptance.